# Duration of intervals in the care seeking pathway for lung cancer in Bangladesh: A journey from symptoms triggering consultation to receipt of treatment

**Adnan Ansar** [1,2]*, **Virginia Lewis**[1,3], **Christine Faye McDonald**[2,4,5], **Chaojie Liu** [6], **Muhammad Aziz Rahman**[2,3,7,8,9]

**1** School of Nursing and Midwifery, College of Science Health and Engineering, La Trobe University, Melbourne, Australia, **2** Institute for Breathing and Sleep (IBAS), Melbourne, Australia, **3** Australian Institute for Primary Care and Aging, La Trobe University, Melbourne, Australia, **4** Department of Respiratory & Sleep Medicine, Austin Health, Melbourne, Australia, **5** University of Melbourne, Melbourne, Australia, **6** School of Psychology and Public Health, La Trobe University, Melbourne, Australia, **7** School of Health, Federation University Australia, Berwick, Australia, **8** Department of Noncommunicable Diseases, Bangladesh University of Health Sciences (BUHS), Dhaka, Bangladesh, **9** Faculty of Public Health, Universitas Airlangga, Surabaya, Indonesia

* dr.adnan.ansar@gmail.com, ansar.a@students.latrobe.edu.au

**Data Availability Statement:** The data that support this study cannot be shared publicly because of privacy issues; data include identifiable personal

## Abstract

Timeliness in seeking care is critical for lung cancer patients' survival and better prognosis. The care seeking trajectory of patients with lung cancer in Bangladesh has not been explored, despite the differences in health systems and structures compared to high income countries. This study investigated the symptoms triggering healthcare seeking, preferred healthcare providers (including informal healthcare providers such as pharmacy retailers, village doctors, and "traditional healers"), and the duration of intervals in the lung cancer care pathway of patients in Bangladesh. A cross-sectional study was conducted in three tertiary care hospitals in Bangladesh among diagnosed lung cancer patients through face-to-face interview and medical record review. Time intervals from onset of symptom and care seeking events were calculated and compared between those who sought initial care from different providers using Wilcoxon rank sum tests. Among 418 study participants, the majority (90%) of whom were males, with a mean age of 57 ±9.86 years, cough and chest pain were the most common (23%) combination of symptoms triggering healthcare seeking. About two-thirds of the total respondents (60%) went to informal healthcare providers as their first point of contact. Living in rural areas, lower levels of education and lower income were associated with seeking care from such providers. The median duration between onset of symptom to confirmation of diagnosis was 121 days, between confirmation of diagnosis and initiation of treatment was 22 days, and between onset of symptom and initiation of treatment was 151 days. Pre-diagnosis durations were longer for those who had sought initial care from an informal provider (p<0.05). Time to first contact with a health provider was shorter in this study compared to other developed and developing countries but utilizing informal healthcare providers caused delays in diagnosis and initiation of treatment.

health information. Data files will be available upon reasonable request to Professor Virginia Lewis and on the condition that the La Trobe University ethics committee grants permission for researchers who meet the criteria for access to confidential data. Contact information: V.Lewis@latrobe.edu.au or directly to Human Research Ethics Committee: humanethics@latrobe.edu.au.

**Funding:** The authors received no specific funding for this work.

**Competing interests:** The authors have declared that no competing interest exist.

Encouraging people to seek care from a formal healthcare provider may reduce the overall duration of the care seeking pathway.

## Introduction

Lung cancer is an aggressive disease with poor survival. As the leading cause of cancer deaths, it has a higher incidence and mortality rate than any other cancers globally [1]. According to the World Cancer Report 2020, there were 2.1 million new cases and 1.8 million deaths in 2018 due to lung cancer. Early stage diagnosis of lung cancer may allow surgical resection, the most effective treatment, with the five-year survival rate being as high as 50–70% [2–4] if diagnosed at Stage I. However, around 75% of people present at an advanced stage of their disease [5], where the overall survival is only 10–20% [1] and 1–5% if diagnosed at Stage IV [2–4].

The importance of timely diagnosis and treatment of lung cancer, and the need for indicators to measure timeliness of care are increasingly discussed in the literature [6, 7]. Guidelines for lung cancer management include recommended timeframes for the intervals from one timepoint to another in the disease care pathway [8]. Clinical guidelines in some western countries recommend start of treatment (surgery, chemotherapy or radiotherapy) within two to eight weeks of diagnosis [9–12]. However, these guidelines are developed based on the strength of resourceful health systems of High Income Countries (HICs) [8] and are hard, if not impossible to implement effectively in Low and Middle Income Countries (LMICs) [13]. There is no existing guideline for lung cancer management in most LMICs, including South Asian countries [14].

Despite consensus around best practice for diagnosis and treatment expressed in guidelines, time from the onset of symptoms to diagnosis, on which prognosis is highly dependent, appears to vary widely in practice, as does time to start treatment. Most studies have been conducted in western countries and a major focus of existing studies is about the waiting time between diagnosis and treatment in line with the clinical guidelines. There is a high degree of variability in the patient journey prior to diagnosis according to the existing literature; however, little is known about LMICs. A study in India [15] found similar results from onset of symptom to initiation of treatment in comparison with those conducted in the western countries [16], but there have been no studies conducted in Bangladesh. Bangladesh is one of the most densely populated LMICs with a population of 163 million people. There were over 250,000 premature deaths from noncommunicable diseases in 2018, 26% of which were due to cancer (108,000) [17]. Lung cancer is the most common cancer diagnosis in males in Bangladesh and is the commonest type of cancer-related death [18]. Most cases of lung cancer are diagnosed at an advanced stage when the options for treatment are limited [19].

The health system of Bangladesh is structured quite differently from those in western countries which have informed current understanding of the lung cancer patient journey. There is no structured referral system from primary care to specialist care and there are no restrictions on patients seeking care from specialists without having an appropriate referral from primary care providers. Referral pathways are fragmented and there is no mechanism to know whether consumers have taken up referrals [20]. Doctors with Bachelor of Medicine and Surgery (MBBS) degrees are considered to be general practitioners in Bangladesh. Secondary and tertiary healthcare service is provided in hospitals, which can be public or private services run by non-government organisations and not-for-profit organizations. General physicians and specialist doctors can be accessed publicly or privately in those settings. There are 37 specialized

hospitals (15 public and 22 private) providing regular cancer treatment in Bangladesh, and an estimated 1.7 radiation oncologists per 10,000 cancer patients [17].

In LMICs, informal healthcare providers who lack recognised training comprise a significant portion of the health system, operating generally as individuals rather than in institutions at a primary care level [21]. Pharmacies, "traditional healers" (ayurvedic, homeopathic, unanie/kabiraji—herbal medicine providers) and village doctors are the most common informal healthcare providers in Bangladesh [22]. Pharmacies, which might be more accurately described as retail medicine shops or drug sellers, are often the first point of contact for people in Bangladesh, particularly those from lower socioeconomic status [23]. These pharmacies provide symptomatic treatment and advice on diseases and health conditions, and dispense medication without a prescription, despite this being prohibited under the drug licence law [24].

There is an overall lack of evidence in Bangladesh and across the subcontinent regarding timeliness of seeking care of lung cancer patients. The aim of this study was to measure the intervals between timepoints in the care-seeking pathway of patients with lung cancer in Bangladesh. Sub-questions were (1) do durations differ according to the symptom(s) that motivated patients to seek healthcare [triggering symptom(s)], (2) do durations differ according to which healthcare providers the patients chose to seek initial healthcare from.

## Materials and methods

### Study design and population

A cross-sectional study was conducted in three study sites in Dhaka, Bangladesh. The sites were purposely selected considering variations in patient profiles, and included one public hospital, one university teaching hospital and one low cost philanthropic cancer hospital. The National Institute of Cancer Research and Hospital (NICRH) is a 300-bed tertiary level public hospital dedicated only for cancer care and provides the maximum load of cancer management in the country. Bangabandhu Sheikh Mujib Medical University (BSMMU) is a postgraduate medical institute with a limited 36-bed inpatient capacity oncology department which accepts both public and private referrals. Ahsania Mission Cancer & General Hospital (AMCGH) is the largest non-profit hospital in the country, currently having a 250-bed inpatient capacity dedicated for cancer care. Patients from across the country receive care from these three hospitals.

Inpatients aged ≥18 years with a known diagnosis of lung cancer, who were able to understand the Participant Information and Consent Form and could nominate a family member or carer, if needed, to provide information on their behalf were eligible to participate in the study. Potential participants who were inpatients on the data collection date were identified by doctors or nurses at the study sites from the inpatient files as per the inclusion criteria, after which the investigators approached admitted consecutively eligible participants for written informed consent. Eligible participants were given information about the voluntary nature of their involvement in the study, details of the study and contact details of the investigators. The participant information was read to those participants who were illiterate. If the patient was feeling unwell and did not wish to provide answers themselves, patients were asked if they wanted to nominate caregivers to provide data on their behalf, and with consent from patients, caregivers were interviewed separately. Participants provided a signature as a sign of their consent or thumb impression for those who were illiterate. There was no compensation provided for participating in the study.

## Questionnaire and data collection procedure

A questionnaire was developed, informed by relevant literature about delays in cancer care [25, 26], national surveys and knowledge of the local health system. Questions and response options were designed to be appropriate to the country context and study objectives. The questionnaire was developed in English and then translated into Bengali with the whole process going through 'translation and back translation' involving two of the investigators and two public health researchers, all of whom were bilingual. The Bengali version of the questionnaire was pilot-tested to ascertain consistency, appropriateness of language, and the best sequence of questions. The questionnaire was pilot-tested with eight lung cancer patients to ensure the questions were understandable and interpreted in a similar way by the participants. Based on the pilot test, several questions were rephrased for better understanding. Participants in the pilot testing were excluded from the final sample for data analysis. The questionnaire collected data under four sections 1) Socio-demographics such as age, sex, area of residence, education, marital status, type of family, and monthly household income, 2) History of illness (comorbidities) 3) Symptoms information including date of onset of symptoms, first symptoms, symptoms that triggered healthcare seeking, 4) Help-seeking practice including choice of healthcare provider for first contact, time to travel to healthcare provider, whether additional healthcare providers were consulted before diagnosis, date of diagnosis, time to travel to diagnostic facility, date of referral for treatment, additional healthcare provider consulted after diagnosis before starting treatment, and date of initiation of treatment. Participants' area of residence was categorised as urban or rural based on the presence of municipal services in the public administrative area.

A total of 418 questionnaires were completed between 10 October 2019 and 13 February 2020, of which 318 were with the patients and 100 were with the patients' nominated caregivers. Of 424 eligible patients, only six refused to participate, with 285 participants from NICRH, 27 from BSMMU and 106 from AMCGH were included in the study. Our intention was to calculate the mean and median for the total lung cancer population admitted to the three hospitals. A sample size of 418 patients enabled a margin of error of less than 1 day in the calculation of the shortest duration indicator (an average of 4 days from diagnosis to referral for treatment) and less than 14 days in calculating the longest duration indicator (an average of 186 days for the entire journey) with a 95% confidence level [27]. The structured questionnaire was administered face-to-face by three data collectors, who were medical doctors, and none was involved in providing treatment to the participants. Two were trained to conduct the survey by the first author who also participated in data collection. Information on different timepoints in the healthcare seeking pathway was collected from hospital clinical files, and from the patient-held clinical files to minimise recall bias. Permission was obtained from the hospital authority and informed consent was sought from the patients to examine both types of clinical files. After obtaining informed consent, a participant's interview was done at the bedside if privacy could be maintained and, in the case of the caregiver, the interview was conducted in the duty doctor's room in private. In addition to asking about different timepoints of care seeking in the disease cycle as part of the questionnaire, data were extracted from patient-held clinical files available at the time of interview. In Bangladesh, patients keep a hard-copy record of their previous treatment documents which includes previous prescriptions, investigation reports and discharge summary of previous hospital admission. They generally have this with them when receiving care in a hospital. This process served as cross-checking to minimise recall bias. Where there was a mismatch of information provided by the participant and that extracted from hospital documents, the dates derived from the hospital were accepted. To ensure data quality, the first author monitored administration of the questionnaire throughout

the data collection period. Five percent of the collected data were reviewed to check the accuracy and completeness of the data. No discrepancies were found in the data.

## Measurement of healthcare seeking intervals

**Date variables.** (i) date of first symptom, (ii) date of first contact with any healthcare provider, (iii) date of diagnosis, (iv) date of referral for treatment and (v) date of treatment initiation.

Date of first symptoms was reported by the respondents and clinical files were also checked to confirm where possible. Date of first contact with a healthcare provider included the date to visit any provider. The date of diagnosis is the date of histopathological or radiological confirmation of lung cancer. Date of referral was the date when the patient was referred by a formal healthcare provider to a tertiary healthcare facility to initiate treatment e.g., chemotherapy, radiotherapy or surgery. There was no referral from informal health care providers to tertiary healthcare facilities in our study.

**Time intervals.** The duration of time intervals was calculated in days from one time point to another. The time interval was recorded as '0' days if the starting and end events occurred on the same day. Seven time intervals were calculated which were as follows:

- Date of onset of symptom(s) to the date of the first contact with any healthcare provider

- Date of the first contact with any healthcare provider to the date of the confirmed diagnosis

- Date of the confirmed diagnosis to the date of the referral for treatment

- Date of the referral for treatment to the date of the initiation of treatment

- Date of the onset of symptom(s) to the date of the confirmation of diagnosis

- Date of the confirmation of diagnosis to the date of the initiation of treatment

- Date of onset of symptom(s) to the date of the initiation of treatment

## Data analysis

Descriptive analyses were performed using frequency, mean and standard deviation (SD), median and interquartile range (IQR) depending on the distribution of the data. Comparison of sociodemographic characteristics and type of healthcare providers for first contact (formal and informal healthcare provider) were performed using chi-square analysis; post-hoc comparisons were adjusted using the Bonferroni approach [28]; Wilcoxon Rank Sum tests were done to compare intervals according to the type of healthcare providers first contacted. The data were analysed using SPSS (version 25) and significance was set at p-value <0.05.

## Ethics

This study obtained ethical approval from the University Human Ethics Committee of La Trobe University, Melbourne, Australia (HEC19154); National Institute of Cancer Research & Hospital, Dhaka, Bangladesh (NICRH/Ethics/2019/504); and Ahsania Mission Cancer and General Hospital, Dhaka Bangladesh (DAM/AMCGH/1900-2019-2063).

## Results

The demographic characteristics of the respondents from each site are described in Table 1. The mean age of the sample was 57 years (SD ±9.86 with a range of 25–82 years). The majority of patients were over 55 years old (68%), with less than 10% aged < 45 years. Participants were

**Table 1. Characteristics of the participants (n = 418).**

| | NICRH (n, %) | BSMMU (n, %) | AMCGH (n, %) | Total (n, %) |
|---|---|---|---|---|
| **Age Group (in years)** | | | | |
| <35 | 7 (2.5) | 0 | 1 (0.9) | 8 (1.9) |
| 35–44 | 20 (7) | 4 (14.8) | 5 (4.7) | 29 (6.9) |
| 45–54 | 65 (22.8) | 2 (7.4) | 29 (27.4) | 96 (23) |
| 55–64 | 120 (42.1) | 12 (44.4) | 45 (42.5) | 177 (42.3) |
| ≥65 | 73 (25.6) | 9 (33.3) | 26 (24.5) | 108 (25.8) |
| **Sex** | | | | |
| Male | 260 (91.2) | 23 (85.2) | 92 (86.8) | 375 (89.7) |
| Female | 25 (8.8) | 4 (14.8) | 14 (13.2) | 43 (10.3) |
| **Marital Status** | | | | |
| Married | 266 (93.3) | 25 (92.6) | 97 (91.5) | 388 (92.8) |
| Not Married | 19 (6.7) | 2 (7.4) | 9 (8.5) | 30 (7.2) |
| **Area of Residence** | | | | |
| Urban | 34 (11.9) | 2 (7.4) | 14 (13.2) | 50 (12) |
| Rural | 251 (88.1) | 25 (92.6) | 92 (86.8) | 368 (88) |
| **Family Structure** | | | | |
| Nuclear family | 120 (42.1) | 10 (37.0) | 34 (32.1) | 164 (39.2) |
| Joint/extended family | 165 (57.9) | 17 (63.0) | 72 (67.9) | 254 (60.8) |
| **Highest Education Level** | | | | |
| Illiterate & below primary education | 196 (68.8) | 21 (77.8) | 52 (51.1) | 269 (64.4) |
| Primary | 48 (16.8) | 3 (11.1) | 13 (12.3) | 64 (15.3) |
| Secondary | 24 (8.4) | 1 (3.7) | 14 (13.2) | 39 (9.3) |
| Higher Secondary | 7 (2.5) | 2 (7.4) | 9 (8.5) | 18 (4.3) |
| Bachelor's degree & above qualification | 10 (3.5) | 0 | 18 (17) | 28 (6.7) |
| **Monthly Household Income** | | | | |
| BDT ≤15000 (≤US$176) | 190 (66.7) | 13 (48.1) | 6 (5.7) | 209 (50) |
| BDT 15001–50000 (US$176–588) | 80 (28.1) | 13 (48.1) | 33 (31.1) | 126 (30.1) |
| BDT 50001–100000 (US$588–1176) | 13 (4.6) | 0 | 38 (35.8) | 51 (12.2) |
| BDT ≥100001 (≥US$1176) | 2 (0.7) | 1 (3.7) | 29 (27.4) | 32 (7.7) |
| **Travel time to first healthcare provider** | | | | |
| <10 min | 91 (31.9) | 8 (29.6) | 43 (40.6) | 142 (34) |
| 10 to 30 min | 125 (43.9) | 11 (40.7) | 24 (22.6) | 160 (38.3) |
| 31 to 60 min | 43 (15.1) | 4 (14.8) | 25 (23.6) | 72 (17.2) |
| 61 min or more | 26 (9.1) | 4 (14.8) | 14 (13.2) | 44 (10.5) |

*1 US$ = 85 BDT (Bangladeshi Taka), NICRH—National Institute of Cancer Research and Hospital, BSMMU—Bangabandhu Sheikh Mujib Medical University, AMCGH—Ahsania Mission Cancer and General Hospital

predominantly from rural areas (88%) and the majority of participants were illiterate (64%). The mean number of years of education was 4 years (SD ±5 and range 0–17 years). The median monthly household income was BDT 15500 (USD 182). The lowest quartile had an income of BDT 10000 (US$ 118) and below, whereas the upper quartile had an income of BDT 100000 (US$ 1176) and above. Significant income differences were found across the three study sites ($p<0.001$). We did not explore stage of lung cancer at the time of diagnosis because the information is generally not available in patient files in Bangladesh; similarly, the data on comorbidities were not sufficient to analyse.

**Table 2. First symptoms and triggering symptoms to seek healthcare provider (n = 418).**

| | Symptoms first noticed | | Triggering symptoms | |
|---|---|---|---|---|
| | Frequency* | % | Frequency* | % |
| Cough | 369 | 88.28 | 275 | 65.79 |
| Chest pain | 236 | 56.5 | 162 | 38.76 |
| Shortness of breath | 108 | 25.84 | 68 | 16.27 |
| Haemoptysis | 103 | 24.64 | 62 | 14.83 |
| Lack of appetite | 43 | 10.29 | 19 | 4.55 |
| Hoarseness of voice | 40 | 9.57 | 19 | 4.55 |
| Weight loss | 10 | 2.39 | 2 | 0.48 |
| Fatigue | 6 | 1.44 | 5 | 1.20 |
| Persistent or recurrent infections | 2 | 0.48 | 0 | 0 |
| Other uncommon symptoms | 65 | 15.6 | 51 | 12.20 |

*as multiple triggering symptoms were reported by a single respondent, the total number is more than 418. % are calculated as a proportion of the total sample.

## Symptoms triggering healthcare seeking

Table 2 shows the first symptom(s) experienced and the symptoms that led to respondents seeking healthcare. Half of the participants (226, 54%) mentioned just one triggering symptom. However, 46% of the participants reported a combination of symptoms triggering healthcare seeking. At least two symptoms triggered healthcare-seeking in 34% of participants, while 11% of participants mentioned three or four symptoms as their triggering factor.

Table 3 shows the differences in timeframe in relation to the combination of triggering symptoms at diagnosis. Participants who reported cough and chest pain experienced longer duration to get diagnosed compared with others ($p<0.002$). However, some of those patients had other symptoms in addition to cough and chest pain.

## Healthcare providers

The majority of participants (60%) had contact with an informal healthcare provider (pharmacy 43%, village doctor and traditional healer 17%) as their first point of contact, whilst 22% went to a General Practitioner (GP) and the remainder to other formal healthcare providers (Table 4). Only 8% of the participants sought care from a single provider before undergoing confirmation of a lung cancer diagnosis. The majority (92%) went to multiple healthcare providers before getting confirmation of their lung cancer diagnosis. Two-fifths consulted two additional healthcare providers (42%), one quarter consulted one additional healthcare provider and 81 (19%) respondents consulted three additional healthcare providers pre-diagnosis.

**Table 3. Association with triggering symptoms and duration of intervals (N = 418).**

| | Frequency (%)* | Onset of symptoms to diagnosis (Median, Range) | p-value |
|---|---|---|---|
| Only Cough | 95 (22.72) | 123 (1069) | 0.31 |
| Cough and chest pain | 91 (21.77) | 157 (1426) | 0.002 |
| Cough and shortness of breath | 54 (12.92) | 139 (496) | 0.08 |
| Cough and haemoptysis | 37 (8.85) | 131 (671) | 0.31 |
| Chest pain and shortness of breath | 22 (5.26) | 146 (467) | 0.18 |

*Multiple triggering symptom reported.

**Table 4. Contact with healthcare providers at different timepoints.**

| Healthcare providers | First point of contact | Additional pre-diagnosis | Additional pre-treatment |
|---|---|---|---|
| | Frequency | Frequency | Frequency |
| | N = 418 (%) | N = 383* (%) | N = 96* (%) |
| Public Healthcare | 28 (6.7) | 319 (76.3) | 34 (8.1) |
| Private and NGO Healthcare | 27 (6.5) | 119 (28.5) | 47 (11.2) |
| GPs (MBBS) | 91 (21.8) | 123 (29.4) | 15 (3.6) |
| Specialist Doctors | 22 (5.3) | 200 (47.9) | 35 (8.4) |
| Pharmacy (over the counter) | 179 (42.8) | 7 (1.7) | 0 |
| Village Doctor & Traditional healer | 71 (17) | 17 (4.1) | 5 (1.2) |

* Multiple responses were allowed.

After first contact, most of the additional pre-diagnosis care seeking was with public healthcare providers (76%) and specialists (48%) (Table 4).

Once a diagnosis of lung cancer was confirmed by a formal healthcare provider, almost one-quarter of participants consulted at least one additional healthcare provider before starting treatment. In our study, all of the participants were referred by their GPs to the tertiary hospitals (study sites) to initiate treatment. Private and NGO healthcare providers were consulted by 47 (11% of 418) participants, 35 (8% of 418) consulted additional specialist doctors and 34 (8% of 418) consulted in public healthcare facilities. Notably, consulting with traditional healers and village doctors was less frequent compared to first point of contact (17%), at additional pre-diagnosis consultation (4%) and additional pre-treatment consultation (1%) timepoints. There is also a large drop in seeking additional care from pharmacy at pre-diagnosis and pre-treatment stage (Table 4).

The type of health care provider who was the first point of contact did not differ by age, sex, marital status or family structure of the participants. However, patients from rural areas (61%), illiterate patients (67%) and patients from the lowest income groups (69%) were more likely to choose an informal healthcare provider as the first point of contact ($p < 0.05$). Education level was associated significantly with type of healthcare provider: illiterate patients were more likely to seek informal care compared with those with bachelor and above education ($p < 0.001$, statistically significant after Bonferroni correction). Similarly household income was associated significantly with type of healthcare provider; participants with household income of <BDT 15000 were more likely to seek informal care ($p < 0.001$, significant after Bonferroni correction) compared with those with an income of >BDT 100,000 ($p < 0.002$, significant after Bonferroni correction) (Table 5).

## Duration of intervals in the care seeking pathway

First contact with informal care (in comparison with formal care) was associated with a shorter duration between onset of symptoms and first contact ($p < 0.05$), but longer durations from first contact to diagnosis, from onset of symptoms to diagnosis, and from onset of symptoms to initiation of treatment ($p < 0.05$) (Table 6).

The entire care pathway was segmented in seven intervals. The median time between onset of symptom and initiation of treatment was 151 days. As a segment, the longest duration in the care seeking pathway was the duration from symptom to diagnosis (median 121 days) and the second longest duration was from first contact with any healthcare provider to diagnosis

**Table 5. First healthcare contact according to sociodemographic characteristics (n = 418).**

| Patient characteristics | Formal healthcare provider | Informal healthcare provider | p-value |
|---|---|---|---|
| | (n, %) | (n, %) | |
| **Age Group (in years)** | | | |
| <35 | 3 (37.5) | 5 (62.5) | 0.40 |
| 35–44 | 11 (37.9) | 18 (62.1) | |
| 45–54 | 31 (32.3) | 65 (67.7) | |
| 55–64 | 79 (44.6) | 98 (55.4) | |
| ≥65 | 44 (40.7) | 64 (59.3) | |
| **Sex** | | | |
| Male | 152 (40.5) | 223 (59.5) | 0.40 |
| Female | 16 (37.2) | 27 (62.8) | |
| **Marital Status** | | | |
| Married | 156 (40.2) | 232 (59.8) | 0.57 |
| Not Married | 12 (40.0) | 18 (60.0) | |
| **Family Structure** | | | |
| Nuclear family | 70 (42.7) | 94 (57.3) | 0.23 |
| Joint/extended family | 98 (38.6) | 156 (61.4) | |
| **Area of Residence** | | | |
| Urban | 27 (54.0) | 23 (46.0) | 0.03 |
| Rural | 141 (38.3) | 227 (61.7) | |
| **Highest Education Level** | | | |
| Illiterate & below primary education | 89 (33.1) | 180 (66.9) | <0.001 |
| Primary education | 28 (43.8) | 36 (56.3) | |
| Secondary & higher secondary education | 31 (54.4) | 26 (45.6) | |
| Bachelor's & above qualification | 20 (71.4) | 8 (28.6) | |
| **Monthly Household Income** | | | |
| BDT ≤15000 (≤US$176) | 65 (31.1) | 144 (68.9) | <0.001 |
| BDT 15001–50000 (US$176–588) | 54 (42.9) | 72 (57.1) | |
| BDT 50001–100000 (US$588–1176) | 28 (54.9) | 23 (45.1) | |
| BDT ≥100001 (≥US$1176) | 21 (65.6) | 11 (34.4) | |

* p-value distribution for Chi-square tests of distribution of participants between seeking care from formal and informal healthcare by participant characteristics.

(median 107 days). The shortest duration was observed between diagnosis and referral for treatment (median 3 days) and onset of symptom to first contact with any healthcare provider (median 10 days) (Fig 1).

**Table 6. Duration of intervals (days) in care seeking with first point of contact\*.**

| | Formal healthcare provider | Informal healthcare provider | p-value |
|---|---|---|---|
| | (N = 168, 40%) | (N = 250, 60%) | |
| | Median (Range) | Median (Range) | |
| Duration between onset of symptoms and first contact with provider | 13 (68) | 10 (358) | 0.03 |
| Duration between first contact with provider and diagnosis | 89 (723) | 121 (1434) | <0.001 |
| Duration between onset of symptoms and diagnosis | 104 (1434) | 136 (1081) | <0.001 |
| Total duration from onset of symptoms to initiation of treatment | 131 (1460) | 171 (1095) | <0.001 |

\*p-value for Wilcoxon Rank Sum test. Not all the intervals are related with the first contact; the most relevant intervals were examined.

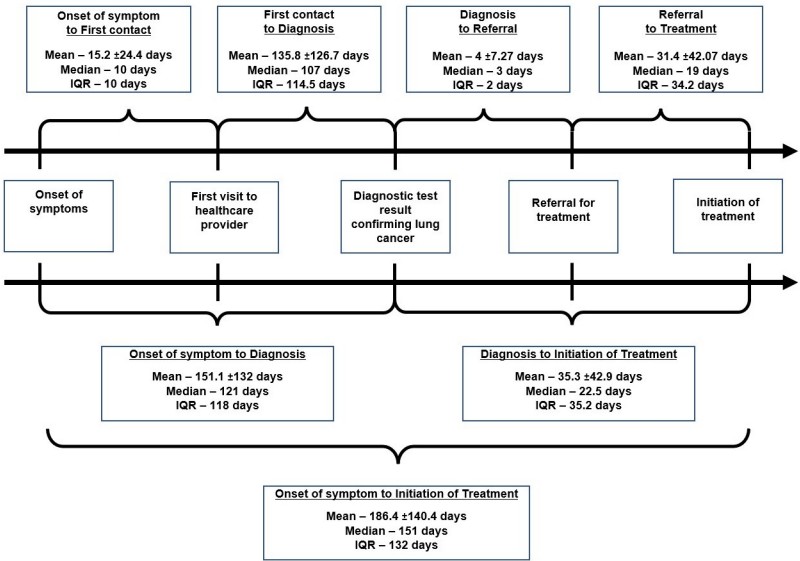

**Fig 1. Timepoints and intervals in the care seeking pathway for lung cancer.**

## Discussion

This study is the first to investigate the care trajectory and duration of intervals in the care seeking pathway for lung cancer in Bangladesh. The focus of this study was on healthcare seeking and symptoms triggered healthcare seeking. In our study, cough was the most frequent first symptom reported by respondents (88%) (Table 2); however, only 23% of the respondents mentioned cough as their only symptom (Table 3) triggering them to seek healthcare or take remedial action. These findings may indicate that not everyone considered a cough as an alarming symptom and potential sign of lung cancer. Apart from the obvious symptom presentation, lung cancer symptoms can be complex in patients with comorbidities [29] and can also be absent from any clear symptom presentation [30]. A history of chronic lung comorbidity and smoking is associated with delayed presentation to a healthcare provider [31]. Patients with lung cancer may also present with a variety of symptoms which are not unique to that condition [32]. The presence of comorbidities with similar symptoms can make the underlying cancer even more difficult to detect [29]. Sometimes, symptoms are misattributed to smoking or to comorbidities such as chronic obstructive pulmonary disease (COPD) or cardiac diseases, resulting in delayed care-seeking [33]. Moreover, stigma [34–36] and fear [34, 37] are associated with delayed care seeking behaviour for lung cancer among smokers which can lead to advanced stage presentation at diagnosis [38].

Historically, smoking rates [39] and lung cancer prevalence [40] are low among women in Bangladesh, which may explain the predominance of males in this study. The mean total duration from onset of symptoms to initiation of treatment for our study was 186.4 ± 140.4 days with a median of 151 days (13–1460 days). This is longer than the duration reported in a study conducted in India, a neighbouring country, where the median duration was 126 days [41]. The total duration in our study, from onset of symptom(s) to initiation of treatment is also longer in comparison to similar studies conducted in western countries such as USA (median of 52 days) [42], Greece (median of 81 days) [43], and Turkey (median 57 days) [16].

Some of the intervals appeared to be shorter than those observed in studies from other countries. The mean duration between onset of symptoms and first care seeking was

15.2 ± 24.4 (median 10) days in this study. This is shorter than time periods reported from symptom onset to contact with a medical doctor in other countries, including 71 days in Nepal [44], 94 days in India [45], and 35 days or 49.9 ± 96.9 days in Turkey [16, 46], however, these studies did not include informal providers as first contact. In this study, informal healthcare providers were the first choice for seeking healthcare reported by most respondents (60%), hence this duration is shorter than other studies mentioned above. Studies conducted in LMICs have reported that, closer vicinity, easier access, and affordability are the main reasons that patients seek care from informal healthcare providers compared to formal healthcare providers [21].

The period from onset of symptoms to confirmed diagnosis was longer than in other countries where similar studies were conducted. In our study the mean duration between onset of symptoms and diagnosis was 151 ± 132 days with a median of 121 days. This is longer compared to studies conducted in India (median of 107 days) [41], Greece (median of 52 days) [43], Turkey (median of 49 days) [16], and Canada (median of 90 days) [47]. As noted above, this difference may reflect the inclusion of informal care providers as the first healthcare provider. This is supported by the finding that the mean duration in our study from first contact with any healthcare provider to diagnosis was longer than other studies (135.8 ± 126.7 days). In Nepal, this step took 50 days on average [44] where the first contact was a medically trained provider. This interval involves the patient reaching qualified doctors and the hospital system, and suggests that the sooner the patient gets diagnosed, the sooner the treatment initiates. In order to ensure early diagnosis, early engagement with a medically trained rather than informal healthcare provider is paramount.

It is noteworthy to mention that the delay occurred in the early phase of the disease care pathway, mostly in the pre-diagnosis phase, and after diagnosis the intervals became shorter compared to pre-diagnosis intervals. The mean interval from diagnosis to initiation of treatment was 35.3 ± 42.9 days with a median of 22 days. Although it was not within the acceptable maximum timeframe for diagnosis to initiation of treatment described in guidelines in Australia, Denmark and Sweden (14 days), it was within an acceptable range for guidelines in the USA (42 days) and the UK (48 days). Further, the duration (diagnosis to initiation of treatment) in this study was shorter than the same interval reported in the USA (median of 27 days [48] and mean of 24.4 ± 54.9 days) and was within the range for medians of 6–45 days reported by Jacobsen et al [49].

The pattern in the duration of intervals between key timepoints in the care seeking pathway for lung cancer in this study suggests that the reason for the duration from onset of symptoms to initiation of treatment was long as care seeking mostly started from informal healthcare in Bangladesh. There were significant differences in timepoints according to whether first contact was with informal healthcare providers compared with formal. While the median duration from onset of symptoms to first contact was significantly shorter for those who used informal providers, intervals between onset of symptoms and diagnosis and treatment were significantly longer for people who sought help from informal providers. Moreover, those who sought care from informal providers were more likely to come from rural areas, had fewer years of education and a lower income. Similar findings were reported in other studies conducted in Bangladesh [50, 51] and India [52]. In addition, traditional healing practice is very widely used in rural Bangladesh, with 75–80% of the rural population dependent on it in the absence of access to formal healthcare [51]. However, only 17% of study participants in this study reported consulting village doctors and traditional healers for management of lung cancer (Table 4). Reliance on traditional healers poses a threat for timely initiation of proper medical care for patients with cancer which is reported in studies conducted in Bangladesh [53], India [54] and

Malaysia [55]. So, while time to seek and receive this care is short, it is unlikely to be effective for someone with lung cancer.

In LMICs like Bangladesh, resource shortage and access barriers are likely to prevent cancer patients from getting timely diagnosis although those who are diagnosed may be treated within the timeline recommended by the clinical guidelines developed in the HICs. This highlights the importance of addressing challenges in the care pathway prior to diagnosis in the LMICs. Unfortunately, there is a lack of contextualised clinical guidelines tailored to the needs of the LMICs. We suspect that there is a possibility that a significant number of people with respiratory diseases, particularly lung cancer, are undiagnosed in Bangladesh's largely rural and disadvantaged population. Exploration of issues of unidentified and unmet need is beyond the scope of this paper. Investment in basic primary health care and use of community health workers could help uncover the huge burden of chronic respiratory disease which is often undiagnosed and unmanaged [56]. While setting up facilities for radiotherapy is a costly option at sub-district levels, investment can be considered for providing incentives to the oncologists who would be interested to work in those settings. Efforts should be given to 'fast-tracking' access to formal healthcare providers for any suspected case of lung cancer for early diagnosis and management. Informal healthcare providers should be trained to recognise lung cancer symptoms and to refer suspected patients early to formal healthcare providers; introducing a reward system could encourage adoption of such practice. Further research will be needed to explore factors associated with longer and shorter durations in the health seeking pathway and their effects on lung cancer survival in Bangladesh.

This study is not devoid of limitations. The date of onset of symptom(s) was collected from patients' or care givers' recollection and could have recall bias, and date of first presentation to a provider is often difficult to ascertain from records in the presence of longstanding comorbidities. This study used data from patients in three urban hospitals in Bangladesh, therefore, it is a limited sample study, so findings cannot represent the experiences of people receiving a diagnosis and treatment for lung cancer in the entire country. However, the included hospitals were the three key referral hospitals for cancer management in Bangladesh and the sample included patients from all over the country and from all levels of socioeconomic status. As this study was conducted in hospital settings there is a possibility of sampling bias; it did not capture the experience of anyone who did not receive treatment from a hospital. A robust study using data from other cancer hospitals in different cities and a country-specific validated tool could increase and ensure generalisability.

## Conclusion

This paper is the first to describe the lung cancer patient journey in Bangladesh. The study suggests that the patterns in the average duration between timepoints in healthcare seeking from onset of symptoms to initiation of treatment in lung cancer care in Bangladesh are different from western and other developing countries, being shorter at some points and longer at others. The findings suggest that the first point of care for most patients was through informal care providers, possibly due to their convenience and ease of access. However, quicker access to informal care may not necessarily translate into quicker provision of lung cancer treatment. In this study, we found that it can lead to significant delays in diagnosis and initiation of treatment. To ensure timely lung cancer diagnosis and care across the whole patient journey, there is a need to better understand the reasons for the intervals between care points observed in this study. This may help to develop health system specific and clinically evidence-based benchmarks for Bangladesh.

## Acknowledgments

We would like to thank the participants, doctors, nurses and data collectors for their time, patience and support. We also thank Professor Dr. Sarwal Alam and Associate Professor Dr. Sadia Sharmin of BSMMU, Professor Dr. Md. Mahbubur Rahman of NICRH, Professor Dr. Qamaruzzaman Chowdhury and Dr. Nahid Sultana of AMCGH for their support in implementing the study in Bangladesh.

## Author Contributions

**Conceptualization:** Adnan Ansar.

**Data curation:** Adnan Ansar.

**Formal analysis:** Adnan Ansar.

**Investigation:** Adnan Ansar.

**Methodology:** Adnan Ansar.

**Project administration:** Adnan Ansar.

**Supervision:** Virginia Lewis, Christine Faye McDonald, Chaojie Liu, Muhammad Aziz Rahman.

**Writing – original draft:** Adnan Ansar.

**Writing – review & editing:** Virginia Lewis, Christine Faye McDonald, Chaojie Liu, Muhammad Aziz Rahman.

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
