## [Decision Letter · Decision Letter 0]

20 Apr 2021

PONE-D-20-39263

Duration of intervals in the care seeking pathway for lung cancer in Bangladesh: a journey from symptoms triggering consultation to receipt of treatment

PLOS ONE

Dear Dr. Ansar,

Thank you for submitting your manuscript to PLOS ONE. After careful consideration, we feel that it has merit but does not fully meet PLOS ONE’s publication criteria as it currently stands. Therefore, we invite you to submit a revised version of the manuscript that addresses the points raised during the review process.

We look forward to receiving your revised manuscript.

Kind regards,

Muhammed Elhadi, MBBCh

Academic Editor

PLOS ONE

Journal Requirements:

2. Please provide further details on sample size and power calculations.

3. In statistical methods, please refer to any post-hoc corrections to correct for multiple comparisons during your statistical analyses. If these were not performed please justify the reasons. Please refer to our statistical reporting guidelines for assistance (https://journals.plos.org/plosone/s/submission-guidelines.#loc-statistical-reporting).

Reviewers' comments:

Reviewer's Responses to Questions

**Comments to the Author**

1. Is the manuscript technically sound, and do the data support the conclusions?

Reviewer #1: Yes

Reviewer #2: Yes

Reviewer #3: Yes

Reviewer #4: Yes

Reviewer #5: Partly

Reviewer #6: Yes

2. Has the statistical analysis been performed appropriately and rigorously? 

Reviewer #1: Yes

Reviewer #2: Yes

Reviewer #3: Yes

Reviewer #4: Yes

Reviewer #5: Yes

Reviewer #6: Yes

3. Have the authors made all data underlying the findings in their manuscript fully available?

Reviewer #1: Yes

Reviewer #2: Yes

Reviewer #3: Yes

Reviewer #4: No

Reviewer #5: Yes

Reviewer #6: No

4. Is the manuscript presented in an intelligible fashion and written in standard English?

Reviewer #1: Yes

Reviewer #2: Yes

Reviewer #3: Yes

Reviewer #4: Yes

Reviewer #5: Yes

Reviewer #6: Yes

5. Review Comments to the Author

Reviewer #1: In this study, Ansar and colleagues evaluate the impact of resource and referral utilization in Bangladesh in duration of wait times between symptom development and treatment of lung cancer. Patients were recruited from three tertiary care hospitals in Dhaka, Bangladesh. They find that rural location, low income, lower educational level and primary self-referral to non-medical providers (such as pharmacists and traditional healers) at the time of symptom onset were all associated with delays in time from symptom onset to diagnosis. Overall, the manuscript is well-written with minor spelling errors, although the introduction is lengthy. Limitations including being descriptive nature, the smaller patient size and limited applicability to analysis (given that it only studies this in Bangladesh, a single low-to-middle income country). These limitations are offset by the specificity of information regarding area of residence, education, and education. Additionally, there is a need for more clear information regarding access to appropriate care in middle to low income countries. However, concerns remain regarding stage at diagnosis, how the time from symptoms to first provider are acquired and concerns about possible bias in attribution of patient behaviors which would need to be addressed.

Critiques:

The introduction is too long, reads like a review article and includes information that could be summarized or is already provided in the discussion. Alternatively, some of this could be briefly summarized in a table under methods.

In p. 6, line 119, the authors refer to “traditional healthcare providers?” I believe the author later refers to these as people who practice holistic medicine (referred to in some places as “traditional healer” or “village doctor”)? If so, the authors should use a consistent label for this group, and possibly place this term in quotes or clearly describe these providers.

p. 10, Time Intervals methods: It is unclear how “estimations” were performed or how much estimation was required (rather than a clear calculation of duration time intervals). Although this may be needed for duration from symptoms to seeking care, this is concerning for recall bias (particularly in other intervals) if based only on patient or family member recollections.

Methods: Definitions of how urban and rural are defined are not clearly stated.

Table 5, p-value for marital status should be 0.98 (decimal omitted)

Table 6: ranges in duration are typically provided in addition to medians. This is addressed for some of these in the discussion, but should be included in the results section rather than the discussion.

The authors do not report the stage of lung cancer at the time of diagnosis. Most lung cancers diagnosed in symptomatic patients are at an advanced stage, but it would be of important to know at what stage(s) these patients were diagnosed in order to compare that to other studies, which may vary in stage at diagnosis (which has been associated with variances in time from diagnosis to treatment, for instance, and for which variations in these durations can variably impact outcomes).

Other spelling errors: p. 18, Line 391, should be “finding.” P. 19, line 395 should be “healers.” P. 19, line 402: should read “symptoms.”

Some of the subjective writing in the discussion and conclusion needs explanation. For instance, in the conclusion, the authors note that “informal care was convenient and easy to access, which…” Were these quotes from patient interviews (qualitative research component), or the writers’ beliefs? If so, they should be presented as hypotheses or potential causes rather than facts. For instance, distrust of established healthcare services and fear of diagnosis have been identified as barriers to care/delays in care in some qualitative research studies.

Avoiding attribution of cause will help with the authors’ later point that “there is a need to better understand the reasons for the intervals…”

The figure needs to be more crisp (did not show up well/was blurry on the provided pdf formatting).

Reviewer #2: The study methodology was sound enough and the article was well written in palatable sentences that can meet the publication criteria in this well reputed journal, PLOS One. The analysis was good as well.

Reviewer #3: This is a very interesting study that is well designed and well written. I have the following comments for the authors to consider that may improve the manuscript.

1. I would be interested in knowing the relationship of the authors (all Australian) with the people they worked with in Bangladesh. I found it somewhat surprising that no Bengali collaborator was included on this manuscript and would consider adding one if possible. This does not necessarily need to be included in the manuscript, although I would like a comment in the response.

2. For those who are not familiar with the term “informal healthcare providers”, please provide explanation or example in the abstract.

3. The wording in the sentence “Although only one fifth the size of its neighbour 92 Pakistan in geographical area, it has just on 75% the population of that country.” Is odd on the bolded section. Perhaps this should read “geographical area, the population is only slightly smaller (75%) that Pakistan’s population”

4. I really appreciated the detailed contest on the Bengali lung cancer statisticis and the health system.

5. I am a little confused about how “traditional healthcare providers” are informal healthcare providers. To me this term indicated traditional hospital-based. I wonder if this is mean to include more of the village doctors? Can you define or clarify this term. I see this is defined later… I might take out the word healthcare or at least move the examples up. Later this is called a traditional healer (which may be a good term)

6. The introduction is very long. It could be shortened by removing some of the non-specific information about lung cancer since there is ample in the discussion of lung cancer in Bangladesh

7. It would be helpful to know how often the providers did or didn’t refer. Was this all consecutive patients? It would be helpful to be able to know a little about potential selection bias. Did all providers refer?

8. Was there any patient compensation?

9. Please comment on the predominance of men in the study.

10. There is a missing ( in line 279.

11. This would be a little easier to read with tables at the end of the manuscript.

12. The discussion is also quite long. It would be nice to consolidate some of the pharmacy discussion, as this is included with references in separate paragraphs.

13. In limitations, please commend on how this sample compares to all people getting a diagnosis. I suspect this is a limited sample, which should be included as a limitation.

Reviewer #4: PONE-D-20-39263

Duration of intervals in the care seeking pathway for lung cancer in Bangladesh: a journey from symptoms triggering consultation to receipt of treatment.

This is a cross-sectional study of 418 patients, majority male, seeking care for lung cancer in Bangladesh. Participants were interviewed in-person and underwent medical record review. Overall, this is extremely well written, logical, and succinct with only a few minor grammatical and typo errors. The majority of participants visited informal healthcare providers first rather than formal healthcare providers (60%). This is incredibly important information as to where people whether in rural or urban settings are getting their healthcare. The authors should be commended in researching a population that is often overlooked due to therapeutic nihilism and that so many were illiterate (64%!). This can make research incredibly difficult, but they took every precaution including informed consent and have documented this incredibly important information. Well done!

Minor comments:

1. Why do the authors think the majority of the population is male? Are women not seeking care? Do they not smoke? Are they not seeking care at all? Mentioning this in the discussion section would be helpful.

2. Line 92: sentence is awkward, would revise. “it has just on 75% the population of that country”. Would suggest: “Bangladesh is geographically much smaller but has 75% of the population of Pakistan.” Or something similar.

3. Line 191 & 208: typo in “minimise” should be: “minimize” but I defer to the editor

4. Line 257: would say “less than or < 45 years” rather than “under 45 years”

5. Line 257: rural areas. The area needs an (s).

6. Line 282: end sentence after cancer diagnosis and remove “of lung cancer”. This is redundant.

7. Table 5: please explain if the p-value is for the distribution of the covariates within the column rather than comparing the urban percentages across the row between formal and information. This can be denoted as a footnote.

8. Line 391: “finding” is misspelled.

9. Line 406: Recognize rather than “recognize” but I defer to the editor

Reviewer #5: This is an interesting paper in an area where there’s little evidence.

I note all of the guidelines quoted in the introduction section. It would be good to see any kind of guidance designed for lower income settings, as it is difficult to apply the standards of very wealthy countries to poorer settings (perhaps something from IARC instead?).

I am not sure how relevant some of the other references are either, for example reference 13 on changes in presenting symptoms – these are very UK specific and are not really applicable to LMIC settings. I would be inclined to restructure the introduction de-emphasise the western country examples – I know there is some LMIC material, but I think there is too much from developed countries.

Reference 21 should be updated, there must be some more recent information on the numbers of hospitals providing cancer treatment in the country.

It would be good to know which literature informed the questionnaire; was it adapted from any existing instrument? Was there any formal validation process?

It might be good also to describe the provision of primary care in Bangladesh – for example, whilst general practitioners are referred to in the paper, this can mean something quite different in LMIC’s. Typically not many medical doctors are classified as general practitioners.

The study draws some interesting comparisons between diagnostic intervals in Bangladesh and other countries – but there are a few methodological issues.

• Date of first symptom can be very difficult to define with lung cancer. It often occurs on the background of chronic respiratory symptoms and the change can be very subtle eg in the nature of the cough or some additional chest pain. Sometimes there is little change at all in symptoms which may have been present for 20 years or more. This should be highlighted in the discussion section along with comments on the validity of the self-reports used

• Date of first presentation can also be difficult to define in the context of multiple or long lasting symptoms and there is often a discrepancy between what patients say and primary care records. It may be helpful to refer to some of the methodological diagnostic intervals work – eg the papers on appraisal intervals by Fiona Walter or the Aarhus statement on diagnostic interval studies. The International Cancer Benchmarking Partnership has included lung cancer in its work and that may be worth referring to as well.

Some other points:

• There should be more emphasis on the huge amount of chronic respiratory disease which goes undiagnosed in Bangladesh. While I appreciate the authors acknowledge this limitation, there should be greater recognition that much lung cancer remains undiagnosed, particularly in poor, rural and remote populations in South Asia.

• I note the recommendation for awareness raising campaigns. I am not sure how meaningful such recommendations would be in the context of the study. As partly highlighted in the paper, there are enormous financial, cultural and social barriers to presenting with lung cancer symptoms in LMIC’s.

• Some authors describe the interval between first noticing a symptom and reporting it, even reporting it to a family member; diagnostic journeys can be rather convoluted and certainly not linear. There is no mention in the paper of stigma, yet this has been recognised as an important barrier to presenting with symptoms that may indicate cancer in LMIC’s.

• It would be good to see some recommendations which are a little bit more tailored to the local financially constrained environment. For example, investment in basic primary health care and use of community health workers can help uncover the huge burden of chronic respiratory disease which is often undiagnosed and unmanaged. Any programme targeting lung cancer would need to be undertaken within this broader context of chronic respiratory disease. Treatments for lung cancer are expensive as are diagnostic investigations such as CT scanning and bronchoscopy. It would be better to ground any recommendations for improved diagnosis of lung cancer in the reality of the current capacity in Bangladesh. What kind of investment might achieve the most in settings where treatments such as chemotherapy, radiotherapy and thoracic surgery are severely limited?

The paper is, nevertheless, of interest because we have very little information to date on cancer diagnostic intervals in South Asian countries and this would be an important addition. I think if it could be better grounded in the international diagnostic interval literature, and if more attention could be paid to local health system and individual factors which might limit any improvements, the paper would be considerably improved.

Reviewer #6: This is an interesting paper on the care seeking pathway for lung cancer in Bangladesh, a subject that had not been studied yet. It shows that the time between the first contact with a healthcare provider and the diagnosis of lung cancer is quite long. This seems to be related to the fact that many patients contact an informal healthcare provider, mainly a pharmacy, before they contact a formally trained healthcare provider. This finding is important as timely diagnosis is crucial in lung cancer. Creating awareness of lung cancer symptoms and encouraging people to seek care from a formal healthcare provider may reduce the overall duration of the care seeking pathway.

My main comments:

Abstract:

As the care seeking trajectory of lung cancer has been studied extensively, please explain why a study Bangladesh is needed.

Please specify “other countries” in the conclusion, are these developing countries or western countries?

Introduction:

The introduction is quite lengthy. Some suggestions to shorten it:

- Please shorten the description of the recommended timeframes in guidelines. Maybe just describe the range or mean for the different time intervals.

- The part about presenting symptoms seems to be a bit out of place and could be left out, as the main presenting symptoms of lung cancer are well-known. Unless you have a hypothesis why presenting symptoms would be different in Bangladesh? Then you should also compare them to the literature.

- I found it very interesting to read about the health system in Bangladesh, it definitely helps to put the results in perspective, so this should remain part of the paper. However, details not directly relevant to the research question, such as e.g. the population of Pakistan, could be left out. Besides, the number of hospitals is based on an article from 2013, maybe more recent data are available?

- As to the research questions, I am missing the ‘why’ for the first two. Why is it important to look at the symptoms that motivated patients to seek care and which healthcare providers they visited? What problem could it solve? I would say your main question is the timeline of the care seeking pathway. A subquestion would then be whether some symptoms lead to a longer care seeking pathway as raising awareness about these symptoms could possibly shorten this time. Another subquestion would be for which health care providers the timeline is longer and then find ways to improve this.

Methods:

- Page 8, line 189: A part of questionnaire is about “history of illness”, what exactly does this mean? comorbidities?

- Page 10, line 218: Please further describe how the different time-points were determined. E.g:.

-date of first symptoms: was this as reported by the patient in the questionnaire? Based on the medical file?

- date of first contact with a health care provider: also including traditional health care providers and pharmacy?

- date of diagnosis of lung cancer: either histological or radiological?

- date of referral for treatment: by which health care providers and to whom, which treatments included? I assume surgery/radiotherapy/chemotherapy?

- date of treatment initiation, which treatments included?

- Page 10, line 222: Instead of the long text explaining the different time intervals, I would rather suggest a graph.

Results:

- Please use subheadings to structure the results.

- Page 11, line 253: Very impressive inclusion rate! Do you know the reason for refusal in the 6 patients?

- Table 1: What is the reason not to test for differences between locations in patient characteristics?

- Table 3: This table could be left out, as it does not add very much to the whole story. Rather, it could be interesting to add whether the timeframes differ by symptoms at diagnosis.

- Page 14, line 295: It is mentioned that the additional health care professionals visited at pre-diagnosis / pre-treatment were much less often traditional healers/village doctors, but there is also a large drop in pharmacy. This should be mentioned.

- Page 14, line 298: Rather describe something like: The type of health care provider who was the first point of contact did not differ by age, sex, marital status or family structure. However, patients from rural areas, illiterate patients and patients from the lowest income groups were more likely to choose an informal healthcare provider as the first point of contact.

Table 6: I assume the numbers in the table are days? Please add the number of patients in the header of the two columns?

A lot of different time intervals are discussed. I would prefer to only discuss in the text the 4 time intervals between the 5 time points and not the other overlapping time intervals. Just mentioning them in the figure or table seems enough to me

Discussion:

Were informal care providers also included in the other studies that you cite? This could explain the shorter duration between first onset of symptoms and first contact with a healthcare provider.

The comparison with existing literature is quite extensive and could be shortened, while the strengths and weaknesses of the study are hardly discussed and could be given more attention.

6. PLOS authors have the option to publish the peer review history of their article (what does this mean?). If published, this will include your full peer review and any attached files.

Reviewer #1: No

Reviewer #2: **Yes: **Shahjada Selim

Reviewer #3: No

Reviewer #4: No

Reviewer #5: No

Reviewer #6: No

---

## [Author Response · Author response to Decision Letter 0]

6 Jul 2021

PONE-D-20-39263 Response to reviewers’ comments provided as an attachment which contains all the responses in a table.

---

## [Decision Letter · Decision Letter 1]

3 Aug 2021

PONE-D-20-39263R1

Duration of intervals in the care seeking pathway for lung cancer in Bangladesh: a journey from symptoms triggering consultation to receipt of treatment

PLOS ONE

Dear Dr. Ansar,

Thank you for submitting your manuscript to PLOS ONE. After careful consideration, we feel that it has merit but does not fully meet PLOS ONE’s publication criteria as it currently stands. Therefore, we invite you to submit a revised version of the manuscript that addresses the points raised during the review process.

We look forward to receiving your revised manuscript.

Kind regards,

Muhammed Elhadi, MBBCh

Academic Editor

PLOS ONE

Journal Requirements:

Reviewers' comments:

Reviewer's Responses to Questions

**Comments to the Author**

1. If the authors have adequately addressed your comments raised in a previous round of review and you feel that this manuscript is now acceptable for publication, you may indicate that here to bypass the “Comments to the Author” section, enter your conflict of interest statement in the “Confidential to Editor” section, and submit your "Accept" recommendation.

Reviewer #1: (No Response)

Reviewer #2: All comments have been addressed

Reviewer #3: All comments have been addressed

Reviewer #5: All comments have been addressed

Reviewer #6: All comments have been addressed

2. Is the manuscript technically sound, and do the data support the conclusions?

Reviewer #1: Yes

Reviewer #2: Yes

Reviewer #3: Yes

Reviewer #5: (No Response)

Reviewer #6: Yes

3. Has the statistical analysis been performed appropriately and rigorously? 

Reviewer #1: Yes

Reviewer #2: Yes

Reviewer #3: Yes

Reviewer #5: (No Response)

Reviewer #6: Yes

4. Have the authors made all data underlying the findings in their manuscript fully available?

Reviewer #1: No

Reviewer #2: Yes

Reviewer #3: Yes

Reviewer #5: (No Response)

Reviewer #6: No

5. Is the manuscript presented in an intelligible fashion and written in standard English?

Reviewer #1: Yes

Reviewer #2: Yes

Reviewer #3: Yes

Reviewer #5: (No Response)

Reviewer #6: Yes

6. Review Comments to the Author

Reviewer #1: The authors present a study of provider and referral utilization in Bangladesh and the impact on wait times between symptoms to diagnosis and ultimately treatment. The authors' have adequately responded to the reviewers' critiques, leading to improvements in the manuscript clarity and impact. Minor concerns remain as detailed below, one of which includes a concern about the provided Figure 1:

p.13, line 292- In Table 1, it appears the upper quartile had a monthly income of BDT >/= 100,001, but the text states this upper quartile was had monthly income BDT 50,000. Could the authors please correct or clarify?

p. 23, line 507: The authors cannot directly suggest that informal care was “convenient and easy to access,” as this study did not evaluate reasons for self-referral to informal care providers. At most, the authors can highlight that most patients’ first point of care is through informal care providers, possibly due to convenience and ease of access (as previously described). The authors can also point out that patients were seen more quickly after initiation of symptoms when the first point was care was an informal care provider.

P23, line 511: Do the authors want to highlight that informal care led to significant delays in lung cancer diagnosis and initiation of treatment (not just initiation of treatment alone as it is currently written)?

Figure 1: Unclear why data for “Onset of symptom to diagnosis” is included twice in this figure. Should the bottom value be “Onset of symptom to Treatment”?

Reviewer #2: Well written article of a good study. In future other studies may include many more centre to describe more details of it.

Reviewer #3: Thank you for addressing comments, manuscript is improved. I appreciated the improved flow and detail within the manuscript.

Reviewer #5: (No Response)

Reviewer #6: (No Response)

7. PLOS authors have the option to publish the peer review history of their article (what does this mean?). If published, this will include your full peer review and any attached files.

Reviewer #1: No

Reviewer #2: **Yes: **Shahjada Selim

Reviewer #3: No

Reviewer #5: No

Reviewer #6: No

---

## [Author Response · Author response to Decision Letter 1]

5 Aug 2021

A rebuttal letter responding to each point raised by the academic editor and reviewer(s) uploaded (labelled as 'Response to Reviewers').

---

## [Decision Letter · Decision Letter 2]

31 Aug 2021

Duration of intervals in the care seeking pathway for lung cancer in Bangladesh: a journey from symptoms triggering consultation to receipt of treatment

PONE-D-20-39263R2

Dear Dr. Ansar,

We’re pleased to inform you that your manuscript has been judged scientifically suitable for publication and will be formally accepted for publication once it meets all outstanding technical requirements.

Kind regards,

Muhammed Elhadi, MBBCh

Academic Editor

PLOS ONE

Additional Editor Comments (optional):

Reviewers' comments:

Reviewer's Responses to Questions

**Comments to the Author**

1. If the authors have adequately addressed your comments raised in a previous round of review and you feel that this manuscript is now acceptable for publication, you may indicate that here to bypass the “Comments to the Author” section, enter your conflict of interest statement in the “Confidential to Editor” section, and submit your "Accept" recommendation.

Reviewer #1: All comments have been addressed

Reviewer #3: All comments have been addressed

Reviewer #5: (No Response)

Reviewer #6: All comments have been addressed

2. Is the manuscript technically sound, and do the data support the conclusions?

Reviewer #1: (No Response)

Reviewer #3: Yes

Reviewer #5: (No Response)

Reviewer #6: Yes

3. Has the statistical analysis been performed appropriately and rigorously? 

Reviewer #1: (No Response)

Reviewer #3: Yes

Reviewer #5: (No Response)

Reviewer #6: Yes

4. Have the authors made all data underlying the findings in their manuscript fully available?

Reviewer #1: (No Response)

Reviewer #3: Yes

Reviewer #5: (No Response)

Reviewer #6: No

5. Is the manuscript presented in an intelligible fashion and written in standard English?

Reviewer #1: (No Response)

Reviewer #3: Yes

Reviewer #5: (No Response)

Reviewer #6: Yes

6. Review Comments to the Author

Reviewer #1: (No Response)

Reviewer #3: Nice work., the manuscript is improved. The authors responded appropriately to the comments that were provided.

Reviewer #5: (No Response)

Reviewer #6: (No Response)

7. PLOS authors have the option to publish the peer review history of their article (what does this mean?). If published, this will include your full peer review and any attached files.

Reviewer #1: No

Reviewer #3: No

Reviewer #5: **Yes: **Professor David Weller

Reviewer #6: No

---

## [Editor Report · Acceptance letter]

2 Sep 2021

PONE-D-20-39263R2 

Duration of intervals in the care seeking pathway for lung cancer in Bangladesh: a journey from symptoms triggering consultation to receipt of treatment  

Dear Dr. Ansar:

I'm pleased to inform you that your manuscript has been deemed suitable for publication in PLOS ONE. Congratulations! Your manuscript is now with our production department. 

Kind regards, 

on behalf of

Dr. Muhammed Elhadi 

Academic Editor

PLOS ONE